# Comprehensive Characterization of Secondary Metabolites from *Colebrookea oppositifolia* (Smith) Leaves from Nepal and Assessment of Cytotoxic Effect and Anti-Nf-κB and AP-1 Activities In Vitro

**DOI:** 10.3390/ijms21144897

**Published:** 2020-07-11

**Authors:** Gregorio Peron, Jan Hošek, Ganga Prasad Phuyal, Dharma Raj Kandel, Rameshwar Adhikari, Stefano Dall’Acqua

**Affiliations:** 1Department of Pharmaceutical and Pharmacological Sciences, University of Padova, Via Marzolo 5, 35131 Padova, Italy; stefano.dallacqua@unipd.it; 2Division of Biologically Active Complexes and Molecular Magnets, Regional Centre of Advanced Technologies and Materials, Faculty of Science, Palacký University in Olomouc, Šlechtitelů 27, 78371 Olomouc, Czech Republic; jan.hosek@upol.cz; 3Research Centre for Applied Science and Technology, Tribhuvan University, Kiritipur, 44613 Kathmandu, Nepal; gangaprasadphuyal@gmail.com (G.P.P.); dharmakandel72@gmail.com (D.R.K.); nepalpolymer@yahoo.com (R.A.)

**Keywords:** *Colebrookea oppositifolia*, secondary metabolites, anti-NF-κB, anti-AP-1, cytotoxic activity

## Abstract

Here we report the comprehensive characterization of the secondary metabolites from the leaves of *Colebrookea oppositifolia* Smith, a species used as medicinal plant in the traditional medicine of Nepal. Phytochemical screening of bioactives was performed using an integrated LC-MS^n^ and high resolution MS (Mass Spectrometry) approach. Forty-three compounds were tentatively identified, mainly aglyconic and glycosilated flavonoids and phenolic acids, as well as other bioactives such as coumarins and terpenes were detected. Furthermore, the NF-κB and AP-1 inhibitory activity of *C. oppositifolia* extract were evaluated, as well as its cytotoxicity against THP-1 cells, in order to assess the potential use of this herb as a source of anti-inflammatory and cytotoxic compounds. The results so far obtained indicate that *C. oppositifolia* leaves extract could significantly reduce the viability of THP-1 cells (IC_50_ = 6.2 ± 1.2 µg/mL), as well as the activation of both NF-κB and AP-1 at the concentration of 2 μg/mL. Our results indicate that Nepalese *C. oppositifolia* is a valuable source of anti-inflammatory and cytotoxic compounds. The phytochemical composition reported here can partially justify the traditional uses of *C. oppositifolia* in Nepal, especially in the treatment of inflammatory diseases, although further research will be needed to assess the full potential of this species.

## 1. Introduction

*Colebrookea oppositifolia* Smith is the only species of the *Colebrookea* genus (Lamiaceae). This plant, commonly known as “Indian squirrel tail”, is widely distributed in the subtropical regions of India, Nepal, Pakistan, Myanmar, Thailand, and China, where it grows in hills and plains at altitudes of 250–1700 m [1]. The aerial part of *C. oppositifolia* consists on a branched shrub up to 1.5–3 m tall, with quadrangular stems and elliptic to ovate-elliptic darkish green leaves, 10–18 cm long × 3.5–7.5 cm large, acuminate, with crenulate to serrulate margins, whitish tomentose beneath. The plant shows tiny white flowers arranged in panicles of 1.5–11 cm-long upright spikes, that look hairy and resemble a squirrel’s tail [2]. In Nepal, *C. oppositifolia* is traditionally known as dhusure, dosul, dhulsu, or dhursuli, and its leaves are used mainly to treat ailments of the ocular region such as cataract, corneal opacity, and keratoconjunctivitis, and as anthelmintic [3,4]. In India and China, this herb has been extensively utilized as a traditional remedy for several other ailments, such as headache, fever, dysentery, peptic ulcer, dermatitis, and wounds [5], although its most frequent use is for the treatment of epilepsy, for which the roots of the plant are considered [6]. Medicinal preparations based on aerial parts of *C. oppositifolia* are used by Dai people of the Yunnan province of China as anti-inflammatory, specifically for the treatment of fractures, traumatic injuries, and rheumatoid arthritis [7]. Nevertheless, studies on *C. oppositifolia* have shown that the different parts of this plant could exert several other pharmacological activities [5]: the whole plant and the roots have been reported as effective as antioxidant [8], anticonvulsant [9], and antiulcer [1] agents, while the leaves have been investigated for their cardio-protective [1], anti-fertility [10], and antibacterial [11] properties. However, to the best of our knowledge, the phytochemical composition of this plant remains scarcely explored, and the majority of these bioactivities have not been attributed to specific compounds yet. Flavonoids such as apigenin, alnustin, mosloflavone [12], and flavonoid glycosides such as negletein-6-*O*-β-d-glucopyranoside and 5,7,2′-trihydroxyflavone-2′-*O*-β-d-glucopyranoside [7] have been identified in the aerial parts, together with the terpenes phytol [1] and (+)-14,15-dinor-9α-hydroxy-cis-labd-11(E)-en-13-one [12]. *C. oppositifolia* is also a source of acteoside [13], a phenylethanoid glycoside that possess antioxidant, antimicrobial, neuroprotective, antitumor, and analgesic properties, and has been studied for its anti-inflammatory activity [14]. Nevertheless, studies on the anti-inflammatory potential of the plant are still lacking, despite its traditional uses. Due to the importance of this medicinal plant in Nepal and due to the relatively scarce literature regarding its phytochemical composition and its biological activities, we selected *C. oppositifolia* to proceed with our studies on the valorization of Nepalese medicinal herbs [15,16,17]. In particular, our aim was that to focus on possible natural sources of bioactive constituents or mixtures targeted on inflammation and/or cancer [15], considering that herbs may offer a safer, and often an equally effective, alternative treatment, especially for long-term utilization [18,19]. In this paper, we reported for the first time the comprehensive characterization of the secondary metabolites extracted from the leaves of *C. oppositifolia* from Nepal, obtained by ultrasound-assisted methanolic extraction of the dried aerial parts. In vitro biological activity of the same extract was also evaluated, and results from anti-NF-κB (Nuclear factor-κB), anti-AP-1 (Activator protein-1), and cytotoxicity assays on human cells are here presented.

## 2. Results

### 2.1. LC-MS Phytochemical Characterization of C. oppositifolia Methanol Extract

The chromatographic analysis of *C. oppositifolia* methanol extract (COME) allowed the detection and tentative identification of 43 chemical constituents (Figure 1). The most representative chemical classes were flavonoids (21 compounds) and phenolic acids (12 compounds), while small amounts of coumarins, triterpenes, and anthocyanins were also identified (Table 1). Detected flavonoids and phenolic acids were quantified using calibration curves built from reference standards, and the results were in alignment with previously published quantitative data on the same plant species [20].

The amount of flavonoids extracted from *C. oppositifolia* was 12.23 ± 0.03 mg/g dry weight (DW), showing a higher content compared to the amounts reported for the same species collected in Pakistan (9.9 and 7.3 mg/g DW for methanol/chloroform and aqueous extracts, respectively) [20]. The sum of the amounts of diosmetin (2), quercetin-*O*-glucoside (13), 6,8-di-C-β-glucopyranosylchrysin (14), myricetin-dihexoside (19), baicalin (21), and anisofolin A (20) accounted for more than 5% of the identified flavonoids (Table 2), and among these, (20) was the most abundant flavonoid derivative in the extract, with an amount of 1.87 mg/g. Considering phenolic acids, the total amount in COME was 32.33 ± 0.51 mg/g DW, and again the result was in agreement with the 32.3 and 25.9 mg/g DW previously reported in methanol/chloroform and aqueous extracts of the same plant collected in Pakistan [20].

The most representative derivative in COME was acteoside (27) (Table 3), also known as verbascoside, a phenylethanoid glycoside that has been previously described as a marker compound of *C. oppositifolia* [13] and as the possible co-responsible of numerous biological activities attributed to the plant [5]. In our extract, compound (27), together with its isomer isoacteoside (28), resulted also the most abundant constituent among those quantified in COME, with a cumulative amount of 24.71 mg/g (27 + 28). The two compounds could not be accurately quantified due to the partial superimposition of their chromatographic peaks, however, considering their peak areas, we could determine that (27) was more abundant than (28), with an amount ratio of 2:1. Another four derivatives sharing similar chemical structure were identified among the most abundant constituents of COME, namely β-oxoacteoside (29), martynoside (30), and β-methoxylverbascoside (31) (Table 1 and Table 3), which are here reported in *C. oppositifolia* for the first time.

The other identified compounds comprised the two coumarins umbelliferone (34) and methoxsalen (35) and the spirostanol neotigogenin (42), also here reported in *C. oppositifolia* for the first time.

### 2.2. Cytotoxicity of COME

The COME showed moderate cytotoxic effect on THP-1-XBlue™-MD2-CD14 cells. The IC_50_ value of the extract was calculated to be 6.2 ± 1.2 µg/mL (95% CI 4.8–10.6 µg/mL) (Figure 2). On the basis of this result, the COME concentration of 2 µg/mL was used for further analysis considered as non-toxic.

### 2.3. Inhibition of NF-κB/AP-1 Activities

To evaluate the anti-inflammatory potential of COME, its ability to attenuate the activity of pro-inflammatory transcription factors NF-κB and AP-1 was measured (Figure 3). COME in the concentration of 2 µg/mL was able to significantly decrease the activity of NF-κB/AP-1 by the factor of 4.8 after LPS stimulation. On the other hand, the commercially available drug prednisone (1 µM) reduced their activity only by the factor of 1.2.

## 3. Discussion

*C. oppositifolia* is widely used in the traditional medicine of South-Eastern Asiatic countries, comprising China, India, and Nepal, to treat a wide spectrum of disorders, ranging from inflammatory diseases to epilepsy [1,12]. In particular, in Nepal the leaves are traditionally used to treat ailments of the ocular region such as cataract, corneal opacity and keratoconjunctivitis, and as anthelmintic [3,4]. Furthermore, the leaves of *C. oppositifolia* have been investigated for their cardio-protective [1], anti-fertility [9], and antibacterial [10] properties. However, despite wide traditional uses of this shrub, only limited studies concerning the assessment of its biological activities in vitro and in vivo, as well as the comprehensive characterization of its secondary metabolites, have been published up until now. Comprehensive phytochemical screening of *C. oppositifolia* could lead not only to the identification and further isolation of potential novel active compounds, but could also allow to explore the molecular pathways that are responsible for the bioactivities. One of the most recent studies regarding the phytochemical characterization of *C. oppositifolia* was published in 2016 by Chinchansure et al. [11], who reported the new diterpene (+)-14,15-dinor-9α-hydroxy-cis-labd-11(E)-en-13-one in the acetone extract of the aerial parts, together with nine known flavonoids and flavone-derivatives such as alnustin, mosloflavone, apigenin, anisofolin A, and apigetrin, and the phenolic acids caffeic acid and forsythoside A. Older studies have reported the characterization of the flavonoids quercetin [43], 5,6,7,4′-tetramethoxyflavone [44], chrysin and landenein [1] from the leaves of *C. oppositifolia*, while 5,6,7-trimethoxyflavone and 5,6,7,4′-tetramethoxyflavone have been characterized from the roots [45]. Moreover, *C. oppositifolia* has been regarded as a rich source of acteoside [5], a phenylethanoid glycoside (verbascoside) with a role in the prevention and treatment of various human diseases and disorders [46]. Nevertheless, to the best of our knowledge, limited information about *C. oppositifolia* from Nepal is available. Our comprehensive phytochemical analysis revealed acteoside and its isomer isoacteoside as the most abundant compounds among the ones identified and quantified in COME, with 24.71 mg/g DW extracted from the leaves. Other three acteoside derivatives were identified and quantified in leaf extract, namely β-oxoacteoside (0.81 mg/g DW), martynoside (0.54 mg/g DW), and β-methoxylverbascoside (0.99 mg/g DW), and are here reported for the first time in *C. oppositifolia*. In the last decade, interests on verbascoside derivatives has been growing because of the increasing evidence showing their role in the prevention and treatment of various human diseases and disorders [46], and due to their apparent low toxicity in mammals, as observed in non-human primate kidney (Vero E6) cells and human erythrocytes [47]. Plants with high concentrations of phenylethanol glycosides have been used in the traditional medicine to treat inflammation and microbial infections [46]. Hence, the high amount of verbascoside derivatives in COME could explain, at least in part, the anti-inflammatory activity observed in vitro. In particular, acteoside has been reported to significantly reduce the expression and activity of both COX and NOS by interfering with the activation of AP-1 [48], and to inhibit the overproduction of pro-inflammatory cytokines in serum and cartilage tissues of osteoarthritic rats by decreasing the activation of NF-κB [49]. AP-1 is a transcriptional regulator of cytokine expression in immune cells and it is involved in the modulation of inflammatory processes, but it has also a role in regulating cell proliferation, neoplastic transformation, and apoptosis. For this reason, AP-1 is considered an interesting target for novel therapeutic applications against inflammation and cancer [50]. On the other hand, NF-κB is a pro-inflammatory transcription factor that plays a determinant role in promoting cellular stress. When activated in the cytosol, NF-κB can activate pro-inflammatory genes by binding to their promoter region, thereby upregulating the production of pro-inflammatory mediators like TNF-α and IL-6 [51]. Also this transcriptional factor is implicated in the regulation of pre-carcinogenic cell transformation, and is considered another interesting target for both anti-inflammatory and anti-tumor agents [52]. Inflammatory response is regulated at several different levels: the gene level (mediated by transcriptional regulation of particular genes), the signal-transduction level (encompassing regulation of intracellular signaling pathways), and, finally, the cellular level (including cell differentiation, activation, and regulation of cell-cell signaling). The main control point has been proposed to lie at the level of transcriptional control [53], hence targeting transcriptional factors such as NF-κB and AP-1 represents a promising therapeutic approach. In our study, we observed that COME, at the concentration of 2 µg/mL, was able to significantly decrease the activity of both NF-κB and AP-1 by the factor of 4.8 after LPS stimulation. This activity factor was four-time higher than that of the commercially available anti-inflammatory drug prednisone used as control, which reduced the activity of both the mediators only by the factor of 1.2, at a concentration of 1 µM. Among the compounds characterized in COME, flavonoids have been largely studied for their anti-inflammatory and anti-tumor activity, and specific members of this family detected in the same extract, namely chrysin, quercetin-*O*-glucoside, and diosmetin, have been reported as NF-κB and AP-1 blockers [54,55,56,57,58]. Also phenolic acids have been already studied for their anti-inflammatory activity, and compounds such as chlorogenic acids, ferulic acid and quinic acid, characterized in COME, have been reported to attenuate the production of pro-inflammatory cytokines in murine cells by down regulating the NF-κB pathway [59,60,61]. Among the other compounds identified in COME, the two coumarins umbelliferone and methoxsalen have been reported to suppress the expression of NF-κB in animal models [62,63], while peonidin-3-glucoside to reduce the metastasis of lung cancer cells by acting via different mechanisms, among which the blocking of AP-1 activation [64]. Overall, these constituents could also contribute to the observed anti-inflammatory and cytotoxic activities of COME.

Considering the results presented in this work, the traditional uses of *C. oppositifolia* in Nepal could be associated to the phytochemical composition of the secondary metabolites of COME. In particular, the presence of relatively high amounts of flavonoids and phenolic acids could explain both the anti-inflammatory properties of the leaves, here confirmed in vitro, as well as the anthelmintic and antibacterial activities, that have been already described for both the classes of compounds [65]. On the other hand, anti-inflammatory potential determined by in vitro anti-NF-κB/AP-1 assay used in this paper could not be obtained by in vivo tests or clinical trials, due to complex and unrevealed pharmacodynamic and pharmacokinetic properties of plant extracts. Transcription factors NF-κB and AP-1 are not only pro-inflammatory, but also contribute to other physiological processes. Hence, the inhibition of NF-κB/AP-1 signaling pathway could lead to undesired side effects [66]. However, if the anti-inflammatory effect of COME is proven in vivo and the particular mechanism of action is elucidated, there will be a therapeutic potential to treat a broad range of inflammatory diseases, at least as food supplement, because of the high popularity of complementary and alternative medicine among the general population [67].

## 4. Materials and Methods 

### 4.1. Plant Collection and Identification

The leaves of *C. oppositifolia* were harvested in May 2017 by G. Prasad and D. Raj in the Bardibas municipality, Mahottari District, Province No. 2, Nepal (26°58′6.8″ N 85°50′42.7″ E). The botanical identification of the plant was confirmed by prof. R.P. Chaudhary, using available literature [2]. The collected specimens were washed in running water and dried in the shadow at room temperature for 7 days, then mounted on an herbarium sheet. A voucher specimen was deposited at the National Herbarium and Plant Laboratories, Godawari-5, Lalitpur, Nepal, under the representative code: MD01 (KATH). 

### 4.2. Extraction of Plant Material

Extraction of dried plant material was performed following a previously published protocol [15]. Briefly, 50 g of dried and grinded leaves were suspended in 250 mL of methanol and extracted at room temperature in an ultrasonic bath (Branson, Danbury, CT, USA) for 2 h, shaking the sample every 30 min to homogenize the mixture. Afterwards, the solid residue was recovered by centrifugation at 4000 rpm for 10 min and further extracted for other 30 min, using an additional 250 mL of methanol as described above. Finally, the liquid extracts were pooled, filtered and dried under vacuum at 40 °C to constant weight, and the dried residue was stored at −20 °C until analysis. The whole extraction procedure was performed in triplicate. The mean percentage yield of the crude COME obtained was 15 ± 1.3%.

### 4.3. LC-MS^n^ and UPLC (Ultra-High Performance Liquid Chromatography)-QTOF (Quadrupole-Time of Flight) Analyses of Secondary Metabolites

Polar constituents of COME were tentatively identified by an integrated LC-MS approach, comparing the fragmentation patterns with literature data and with standard compounds, when available. Further information was obtained searching for possible candidates in freely available MS databases by comparison of the accurate *m/z* values and the calculated molecular formulas, obtained from LC-QTOF analysis. The dried extract was dissolved in methanol at a concentration of 5 mg/mL, and the solution was filtered through a 0.45 µm Millipore filter. A Varian 212 binary pump equipped with a Varian Prostar 430 autosampler and coupled to a Varian 500 Ion Trap mass detector (MS) was employed. The mass spectrometer was equipped with an Electrospray Ionization (ESI) ion source, operating in both negative [ESI(−)] and positive [ESI(+)] ion modes. Stationary and mobile phases, chromatographic conditions and MS parameters have been already reported in [15]. For quantification of flavonoids and phenolic acids, calibration curves of rutin and chlorogenic acid, respectively, were obtained using concentrations ranging from 10 to 120 mg/L. The calibration curves were *y* = 13011*x* – 192439 (R^2^ = 0.992) and *y* = 523050*x* (R^2^ = 0.990) for rutin in ESI(−) and ESI(+), respectively, and *y* = 17133.7*x* + 18842 (R^2^ = 0.997) for chlorogenic acid in ESI(−). Quantitative results were expressed as milligrams per grams of dry weight (DW).

Accurate *m/z* values were obtained using a Waters Acquity H-Class UPLC system coupled to a Waters Xevo G2 QTOF MS detector, operating in both ESI(−) and ESI(+) modes. MS parameters have been already reported in [15], while the chromatographic conditions were kept as previously described.

### 4.4. Cell Viability Determination

The influence of COME on cell viability was determined as previously described [15]. Briefly, cells [THP1-XBlue™-MD2-CD14 (Invivogen, San Diego, CA, USA)] were incubated for 24 h with increasing concentrations (1.25–20 µg/mL) of COME dissolved in DMSO. The solvent did not exceed the concentration of 0.1% (*v/v*). After 24 h incubation, the amount of metabolically active (i.e., live) cells was evaluated by Cell Proliferation Reagent kit WST-1 (Roche Diagnostics, Basel, Switzerland) according to the manufacturer’s manual. Relative cell viability was calculated as the ratio of absorbance values of COME-treated to DMSO-only-treated cells. The IC_50_ value of COME was determined by four parameter logistic (4PL) analysis from the obtained viability curve. For further analysis, the concentration of COME 2 µg/mL was considered as non-toxic.

### 4.5. Determination of NF-κB and AP-1 Activity

THP-1-XBlue™-MD2-CD14 cell line was used as a model for determination of the effect of COME on the activation of transcription factors NF-κB and AP-1 (key pro-inflammatory transcription factors). This genetically modified cell line has been specially designed for testing the activity of these transcription factors. We analyzed the ability of COME to suppress the activity of lipopolysaccharide (LPS)-activated NF-κB/AP-1 as previously described [15]. Briefly, cells were pre-treated with COME at the non-toxic concentration of 2 µg/mL and control drug prednisone 1 µM (Sigma-Aldrich, Milan, Italy) dissolved in DMSO for 1 h. Subsequently, cells were stimulated by LPS isolated from *Escherichia coli* O111:B4 (Sigma-Aldrich) dissolved in serum-free RPMI 1640 medium (1 µg/mL). 24 h later, the activity of NF-κB/AP-1 was determined by Quanti-Blue™ medium (Invivogen, Toulouse, France) spectrophotometrically on FLUOstar Omega Microplate Reader (BMG Labtech, Ortenberg, Germany) at 650 nm, according to manufacturer’s manual. Results were compared with a control incubated only with DMSO and stimulated with LPS (100% NF-κB/AP-1 activity).

### 4.6. Statistical Analysis

All the biological experiments were performed in triplicate. The results are presented as mean values, with the error bars representing the standard error of the mean (SEM). A one-way ANOVA test was used for statistical comparisons, followed by Fisher’s LSD multiple comparison test. GraphPad Prism 8.01 software (GraphPad Software Inc., San Diego, CA, USA) was used to perform the statistical analysis. A value of *p* < 0.05 was considered as statistically significant.

## 5. Conclusions

*C. oppositifolia* is a shrub largely used as medicinal species in Nepal and other Asiatic countries, nevertheless, exhaustive chemical characterizations of the utilized parts are still lacking. Furthermore, despite one of the traditional uses of the leaves of this plant in Nepal and China is for the treatment of inflammatory diseases, no reports on their anti-inflammatory activity has been published up until now, nor in vitro, nor in vivo. In this article, we report for the first time the comprehensive characterization of the secondary metabolites obtained from the leaves of *C. oppositifolia* of Nepalese origin, and the assessment of its anti-NF-κB, anti-AP-1 and cytotoxic activities in human cell lines. Overall, the preliminary data presented here could contribute to rationalize the medicinal use of this plant by the local communities of Nepal, and could be a starting point for the isolation of potential novel anti-inflammatory and anti-tumor agents.

## Figures and Tables

**Figure 1 ijms-21-04897-f001:**
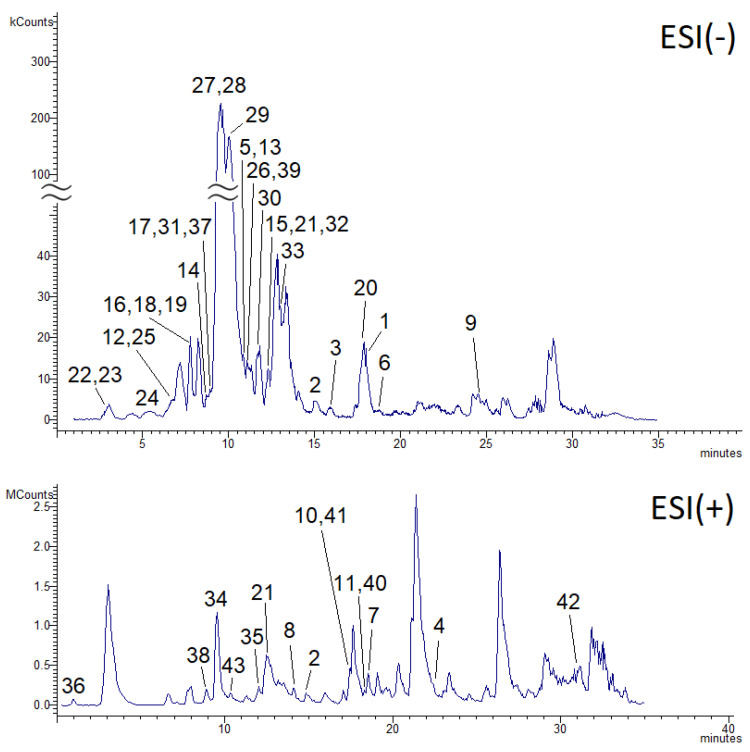
Intensity of base peak chromatograms obtained from the LC-MS^n^ analysis of *Colebrookea oppositifolia* methanol extract (COME) in ESI(−) (upper panel) and ESI(+) (lower panel) modes. The numbers in the chromatograms indicate the tentatively identified phytoconstituents, as reported in Table 1.

**Figure 2 ijms-21-04897-f002:**
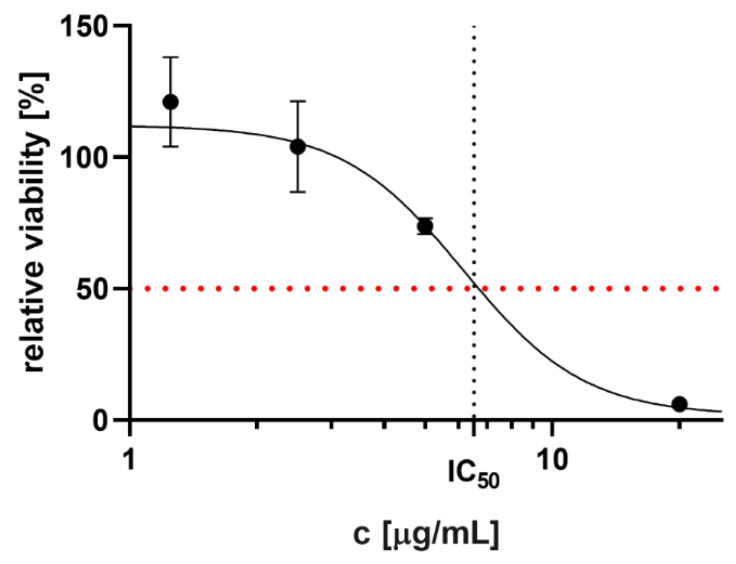
Relative cell viability determined after COME application. Results are expressed as percentage of viability of untreated cells (contained only 0.1 *v*/*v* DMSO), particular point are means ± standard error of the mean (SEM). Dashed lines indicate IC_50_ value.

**Figure 3 ijms-21-04897-f003:**
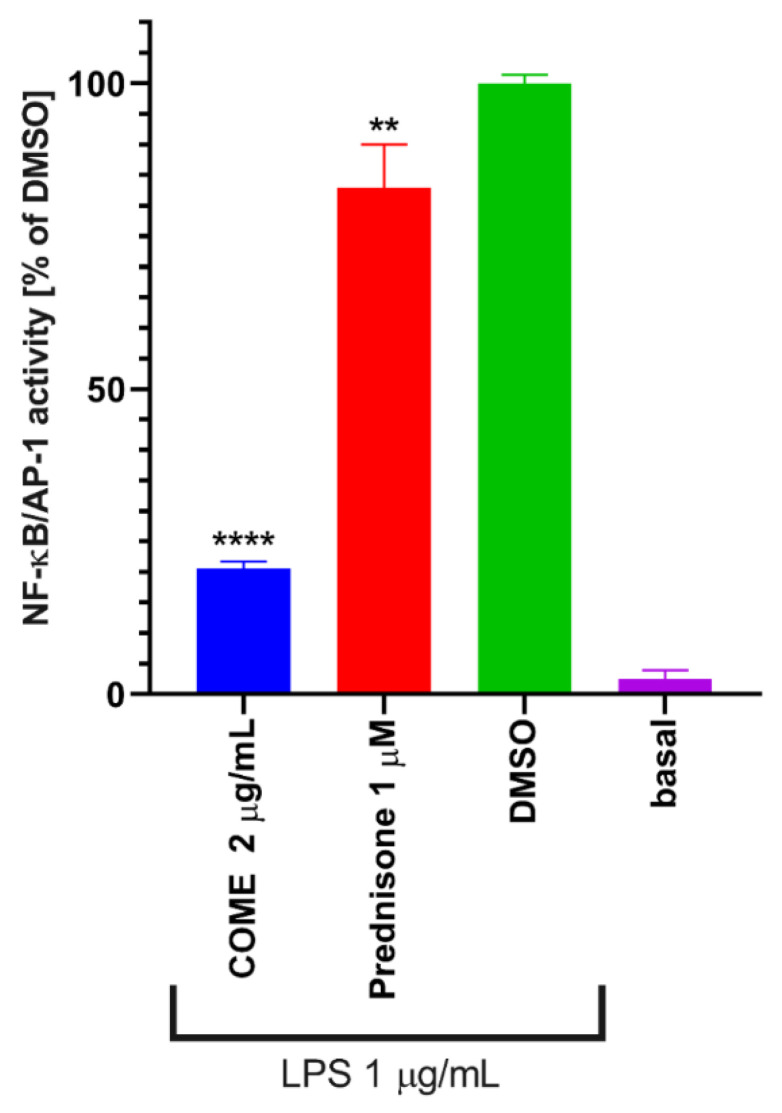
Effect of COME on the activity of NF-κB/AP-1 in LPS-stimulated THP-1-XBlue™-MD2-CD14 cells. Cells were pre-treated in indicated concentrations by COME, prednisone or only solvent (DMSO) for 1 h, after that, LPS was added and NF-κB/AP-1 activity was measured 24 h later. ** indicates statistical significance to positive control (DMSO) *p* < 0.01, **** indicates statistical significance to positive control (DMSO) *p* < 0.0001.

**Table 1 ijms-21-04897-t001:** Identification of polar constituents from the methanol extract of *Colebrookea oppositifolia* by HPLC-MS^n^.

Nº	Experimental *m*/*z*	Theoretical *m*/*z*	ppm	Molecular Formula	ESI(−)/ ESI(+)	MS2 ^#^	MS3	R.T. (min)	Tentative Identification	Chemical Class	Reference
1	253.0494	253.0501	2.9	C_15_H_10_O_4_	−	161 **145** 143 121		18.2	Chrysin	Flavonoid	[12]
2	299.0548	299.0556	2.8	C_16_H_11_O_6_	−	284	256 **227**	14.9	Diosmetin ^a^	Flavonoid	[21]
3	299.0546	299.0556	3.5	C_16_H_11_O_6_	−	284	255 256 **227** 211	15.9	4′-Hydroxywogonin	Flavonoid	[22]
4	299.0909	299.0919	3.5	C_17_H_15_O_5_	+	284	256	23.4	Mosloflavone	Flavonoid	[23]
5	301.0341	301.0348	2.5	C_15_H_9_O_7_	−	**179** 151		10.7	Quercetin *	Flavonoid	-
6	313.0701	313.0712	3.7	C_17_H_13_O_6_	−	298 **299**		19.0	Ladanein	Flavonoid	[12]
7	313.1070	313.1076	2.0	C_18_H_17_O_5_	+	298 283 269 **252**		18.4	Retusin dimethylether	Flavonoid	Metlin
8	331.0824	331.0818	1.9	C_17_H_15_O_7_	+	316	301 **288**	14.1	3,7-dimethoxy-5,3′,4′-trihydroxyflavone	Flavonoid	[24]
9	335.0417	335.0403	4.4	C_15_H_11_O_9_	−	291 272 263 **193**		24.6	6-Hydroxyampelopsin	Flavonoid	[25]
10	345.0961	345.0974	4.0	C_18_H_17_O_7_	+	330 **312** 284		17.5	Eupatorin	Flavonoid	Metlin
11	403.1380	403.1392	3.2	C_21_H_23_O_8_	+	373 355 342 327	358 345 **330** 301	18.1	Irigenin trimethylether	Flavonoid	Metlin
12	461.1073	461.1084	2.5	C_22_H_21_O_11_	−	**299** 301 284		6.9	Diosmetin 7-*O*-glucoside	Flavonoid	[26]
13	463.0885	463.0877	1.8	C_21_H_19_O_12_	−	301		10.6	Quercetin-*O*-glucoside	Flavonoid	[27]
14	577.1544	577.1557	2.4	C_27_H_29_O_14_	−	487 **457** 367 337		8.7	6,8-di-C-β-glucopyranosylchrysin	Flavonoid	[28]
15	593.1500	593.1507	1.3	C_27_H_29_O_15_	−	503 **473** 383 353 325		12.5	Apigenin-6,8-di-*C*-hexoside (vicenin-2)	Flavonoid	[29]
16	595.1651	595.1663	2.1	C_27_H_31_O_15_	−	505 **475** 385 355 271		8.0	Naringenin-6,8-di-*C*-hexoside	Flavonoid	[30]
17	609.1462	609.1456	1.0	C_27_H_29_O_16_	−	301		9.2	Rutin *	Flavonoid	-
18	639.1554	639.1561	1.2	C_28_H_31_O_17_	−	477 **459** 315		8.1	Isorhamnetin-*O*-sophoroside	Flavonoid	[31]
19	641.1334	641.1354	3.3	C_27_H_29_O_18_	−	479 **463** 317		8.0	Myricetin-dihexoside	Flavonoid	[32]
20	723.1698	723.1714	2.4	C_39_H_31_O_14_	−	577 559 453 **269**		17.8	Anisofolin ^a^	Flavonoid	[11]
21	891.1591	891.1620	3.5	C_42_H_35_O_22_	−	**445**	269	12.3	Baicalin (dimer) ^b^	Flavonoid	[33]
22	191.0547	191.0556	5.0	C_7_H_11_O_6_	−	173		3.0	Quinic acid *	Phenolic acid	-
23	193.0499	193.0501	1.1	C_10_H_9_O_4_	-	134		3.2	Ferulic acid *	Phenolic acid	-
24	353.0866	353.0873	2.1	C_16_H1_7_O_9_	-	191 179 **173**		5.6	Chlorogenic acid isomer 1	Phenolic acid	[25]
25	353.0865	353.0873	2.4	C_16_H_17_O_9_	-	191 179 **173**		7.2	Chlorogenic acid isomer 2	Phenolic acid	[25]
26	549.1964	549.1972	1.5	C_27_H_33_O_12_	-	**387** 505 489	225 **207** 163	11.1	12-*O*-(6′-caffeoylhexosyl)jasmonic acid	Phenolic acid	[34]
27	623.1958	623.1976	3.1	C_29_H_35_O_15_	-	**461** 315 297	135	9.8	Acteoside (verbascoside)	Phenolic acid	[35]
28	623.1961	623.1976	2.5	C_29_H_35_O_15_	-	**461** 315 297	135	9.9	Isoacteoside (isoverbascoside)	Phenolic acid	[35]
29	637.1788	637.1769	3.2	C_29_H_33_O_16_	-	475	367 **329** 312	10.4	β-oxoacteoside	Phenolic acid	[35]
30	651.2263	651.2289	4.2	C_31_H_39_O_15_	-	505 **475** 457 329		11.7	Martynoside	Phenolic acid	[35]
31	653.2066	653.2082	2.6	C_30_H_37_O_16_	-	**621** 491 459		9.3	β-methoxylverbascoside	Phenolic acid	[36]
32	951.0719	951.0739	2.2	C_41_H_27_O_27_	-	789 **475**		12.6	HHDP-valoneoyl-glucose isomer 1	Phenolic acid	[37]
33	951.0718	951.0739	2.2	C_41_H_27_O_27_	-	789 **475**		13.3	HHDP-valoneoyl-glucose isomer 2	Phenolic acid	[37]
34	163.0391	163.0395	2.6	C_9_H_7_O_3_	+	145	118 **117**	9.9	Umbelliferone	Coumarin	[38]
35	217.0497	217.0501	1.9	C_12_H_9_O_4_	+	**202** 189 173 161	**174** 146	12.3	Methoxsalen	Coumarin	Metlin
36	132.0810	132.0813	2.4	C_9_H_10_N	+	91		1.0	3-methylindole	Other	Metlin
37	161.0441	161.0450	5.9	C_6_H_9_O_5_	-	117		9.4	Hydroxymethylglutaric acid	Other	[39]
38	201.1020	201.1028	4.2	C_12_H_13_N_2_O	+	160		8.9	Harmalol	Other	Metlin
39	387.1644	387.1655	3.0	C_18_H_27_O_9_	-	225 **207** 163		11.1	12-Hydroxyjasmonic acid glucoside	Other	[40]
40	403.1751	403.1756	1.3	C_22_H_27_O_7_	+	**373** 342		18.3	(7S,8R,7′E)-4-hydroxy-3,5,5′,9′-tetramethoxy-4′,7-epoxy-8,3′-neo-lign-7′-en-9-ol	Other	[41]
41	403.1750	403.1756	1.6	C_22_H_27_O_7_	+	**373** 342		17.9	(7S,8R,7′E)-4-hydroxy-3,5,5′,9′-tetramethoxy-4′,7-epoxy-8,3′-neo-lign-7′-en-9-ol isomer	Other	[41]
42	417.3355	417.3368	3.3	C_27_H_45_O_3_	+	400 **226** 212		31.1	Neotigogenin	Other	Metlin
43	463.1233	463.1240	1.6	C_22_H_23_O_11_	+	301 **258**		10.3	Peonidin-3-glucoside	Other	[42]

^a^ Detected also in ESI(+) mode. *m*/*z*: 301.0705; MS2 fragments: 286, 258. ^b^ Detected also in ESI(+) mode. *m*/*z*: 447.0936; MS2 fragments: 271, 253. * Identification was confirmed by co-injection with reference standard. HHDP—hexahydroxydiphenoyl group. ^#^ Numbers in bold indicate the base peaks.

**Table 2 ijms-21-04897-t002:** Quantitative analysis of flavonoids from *Colebrookea oppositifolia* methanol extract.

ESI(−)/ ESI(+)	Nº	[M-H]^−^/ [M+H]^+^	Compound	mg·g^−1^ DW *
−	1	253	Chrysin	0.53 ± 0.01
	2	299	Diosmetin	0.75 ± 0.02
	3	299	4′-Hydroxywogonin	0.61 ± 0.01
	5	301	Quercetin	0.46 ± 0.00
	6	313	Ladanein	0.53 ± 0.01
	9	335	6-Hydroxyampelopsin	0.54 ± 0.01
	12	461	Diosmetin 7-*O*-glucoside	0.47 ± 0.00
	13	463	Quercetin-*O*-glucoside	0.66 ± 0.00
	14	577	6,8-di-C-β-glucopyranosylchrysin	0.66 ± 0.02
	15	593	Apigenin-6,8-di-C-hexoside	0.54 ± 0.01
	16	595	Naringenin-6,8-di-C-hexoside	0.56 ± 0.00
	17	609	Rutin	0.51 ± 0.01
	18	639	Isorhamnetin-*O*-sophoroside	0.53 ± 0.01
	19	641	Myricetin-dihexoside	0.65 ± 0.01
	20	723	Anisofolin a	1.87 ± 0.01
	21	891	Baicalin (dimer)	0.62 ± 0.00
+	4	299	Mosloflavone	0.51 ± 0.01
	7	313	Retusin dimethylether	0.23 ± 0.01
	8	331	3,7-dimethoxy-5,3′,4′-trihydroxyflavone	0.30 ± 0.02
	10	345	Eupatorin	0.51 ± 0.01
	11	403	Irigenin trimethylether	0.21 ± 0.01
			Total	12.23 ± 0.03

* Values are expressed as mean of results from three analyses ± S.D.

**Table 3 ijms-21-04897-t003:** Quantitative analysis of phenolic acids from *Colebrookea oppositifolia* methanol extract.

Nº	[M-H]^-^	Compound	mg·g^−1^ DW *
22	191	Quinic acid	0.13 ± 0.01
23	193	Ferulic acid	0.02 ± 0.01
24	353	Chlorogenic acid isomer 1	0.74 ± 0.06
25	353	Chlorogenic acid isomer 2	0.10 ± 0.01
26	549	12-*O*-(6′-caffeoylhexosyl)jasmonic acid	0.48 ± 0.08
27 + 28	623	Acteoside + Isoacteoside	24.71 ± 0.10
29	637	β-oxoacteoside	0.81 ± 0.03
30	651	Martynoside	0.54 ± 0.07
31	653	β-methoxylverbascoside	0.99 ± 0.04
32	951	HHDP-valoneoyl-glucose isomer 1	2.17 ± 0.07
33	951	HHDP-valoneoyl-glucose isomer 2	1.64 ± 0.05
		Total	32.33 ± 0.51

* Values are expressed as mean of results from three analyses ± S.D. HHDP—hexahydroxydiphenoyl group.

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
