# Peer review of "Comprehensive Characterization of Secondary Metabolites from *Colebrookea oppositifolia* (Smith) Leaves from Nepal and Assessment of Cytotoxic Effect and Anti-Nf-κB and AP-1 Activities In Vitro"

_ijms, 2020, doi:10.3390/ijms21144897_

Round 1

Reviewer 1 Report

The authors describe in their manuscript i) the phytochemical characterization and ii) the NF-κB and AP-1 inhibitory activity of Colebrookea oppositifolia leaf extracts. Furthermore, correlations between the extract composition and enzyme inhibitory effects were also discussed. These results could be considered to be significant and new. Furthermore, the manuscript is well written and documented.

However, the manuscript needs some revision before publication according to the points 1-10, as follows:

1) Page 2, lines 74, 75: Instead of “methyl extract”, methanol extract should be used.

2) Fig. 1: Compound 26 is missing from Fig. 1.

3) Table 1: compound name “diosmetina” needs to be corrected: diosmetin is the correct name.

4) Table 1: The meaning of the abbreviation “HHDP” in the compounds name “HHDP-valoneoyl-glucose isomer 1 and 2” should be described.

5) Page 6, line 255, Section Materials and Methods: “…COME dissolved in DMSO was tested at final concentrations of 1.25 – 20 μM…” However, in the sections Results and Discussion, concentrations of COME are expressed in μg/mL. Final concentration of COME (in the section Materials and Methods) should also be given in μg/mL.

6) Names of compounds should be consequently written within the Tables and the text.

line 88: “…6,8-di-C-β-glucopyranosyl chrysin…” Tables 1, 2: “…6,8-di-C-β-glucopyranosylchrysin…”   

7) Page 3, line 89: “…baicalin (21)…” However, compound 21 is a dimer of baicalin (as shown in Tables 1 and 2). Consequently, “baicalin (21)” in line 89 needs to be corrected: dimer of baicalin is the correct version.  

8) Page 3, lines 87-89: ‘…Diosmetin (2), 87 quercetin-O-glucoside (13), 6,8-di-C-β-glucopyranosyl chrysin (14), myricetin-dihexoside (19), 88 baicalin (21) and anisofolin A (20) accounted for more than the 5% of the identified flavonoids…” Meaning of this sentence is not unequivocal: The sum of these flavonoids accounted for more than the 5% of the identified flavonoids or each of these compounds (one by one) accounted for more than the 5% of the identified flavonoids? Thus, this sentence should be improved.

9) As shown in the Fig. 1, acteoside (syn. verbascoside) and isoacteoside were not base-line separated. In this special case, their sum amount could only be (accurately) determined and their ratio (as an approximate value) could be given. According to this, Table 3 should be improved.  

10) When reviewing the medicinal significance of acteoside/verbascoside and acteoside-related compounds, their safe use (without side effects) should also be highlighted. Thus, the section Conclusion (after the sentence in the lines 161-163) could be completed with the following (or similar) sentence: Their safe use was also investigated, confirming no cytotoxic side effects on mammalian kidney cells (Vero E6) and human erythrocytes (Zürn et al., 2019 Industrial Crops and Products, https://doi.org/10.1016/j.indcrop.2019.111517).

In conclusion the manuscript deserves publication after a minor revision, taking into consideration the above detailed comments.

Author Response

The authors describe in their manuscript i) the phytochemical characterization and ii) the NF-κB and AP-1 inhibitory activity of Colebrookea oppositifolia leaf extracts. Furthermore, correlations between the extract composition and enzyme inhibitory effects were also discussed. These results could be considered to be significant and new. Furthermore, the manuscript is well written and documented.

Thanks to the reviewer for the positive comment on our work.

However, the manuscript needs some revision before publication according to the points 1-10, as follows:

1) Page 2, lines 74, 75: Instead of “methyl extract”, methanol extract should be used.

The term was changed in the text, as suggested.

2) Fig. 1: Compound 26 is missing from Fig. 1.

Thanks to the reviewer for the observation. The compound was added to the figure.

3) Table 1: compound name “diosmetina” needs to be corrected: diosmetin is the correct name.

The name was corrected, as suggested.

4) Table 1: The meaning of the abbreviation “HHDP” in the compounds name “HHDP-valoneoyl-glucose isomer 1 and 2” should be described.

The meaning of the abbreviation was added to the table.

5) Page 6, line 255, Section Materials and Methods: “…COME dissolved in DMSO was tested at final concentrations of 1.25 – 20 μM…” However, in the sections Results and Discussion, concentrations of COME are expressed in μg/mL. Final concentration of COME (in the section Materials and Methods) should also be given in μg/mL.

We apologize for the mistake. It was a mistype, the unit µg/mL is correct.

6) Names of compounds should be consequently written within the Tables and the text.

line 88: “…6,8-di-C-β-glucopyranosyl chrysin…” Tables 1, 2: “…6,8-di-C-β-glucopyranosylchrysin…”  

The names were corrected, as suggested.

7) Page 3, line 89: “…baicalin (21)…” However, compound 21 is a dimer of baicalin (as shown in Tables 1 and 2). Consequently, “baicalin (21)” in line 89 needs to be corrected: dimer of baicalin is the correct version. 

The dimer of baicalin as reported in Table 1 is an adduct ion formed probably in the ion source during the MS analysis. However, the fragmentation pattern was compared to literature (as reported), and it was matching with that of baicalin (“monomer”). Hence, the compound is present in its natural form (the “monomer”, not dimer) in the extract, but we detected it as an adduct ion. So, the name should not be changed in the text, in our opinion.

8) Page 3, lines 87-89: ‘…Diosmetin (2), 87 quercetin-O-glucoside (13), 6,8-di-C-β-glucopyranosyl chrysin (14), myricetin-dihexoside (19), 88 baicalin (21) and anisofolin A (20) accounted for more than the 5% of the identified flavonoids…” Meaning of this sentence is not unequivocal: The sum of these flavonoids accounted for more than the 5% of the identified flavonoids or each of these compounds (one by one) accounted for more than the 5% of the identified flavonoids? Thus, this sentence should be improved.

The sum of the amounts of these compounds represents the 5% of the identified flavonoids. The sentence was re-formulated as follows: “The sum of the amounts of diosmetin (2), quercetin-O-glucoside (13), 6,8-di-C-β-glucopyranosylchrysin (14), myricetin-dihexoside (19), baicalin (21) and anisofolin A (20) accounted for more than 5% of the identified flavonoids…”.

9) As shown in the Fig. 1, acteoside (syn. verbascoside) and isoacteoside were not base-line separated. In this special case, their sum amount could only be (accurately) determined and their ratio (as an approximate value) could be given. According to this, Table 3 should be improved. 

Thanks to the reviewer for the comment. We corrected the amount in Table 3, showing only the sum of the amounts of the two compounds. Furthermore, in the text we specify that an accurate quantification of the two isomers was not possible due to partial superimposition, however, considering their peak AUCs, we could determine an amount ratio of 2:1 for 27 and 28, respectively.

10) When reviewing the medicinal significance of acteoside/verbascoside and acteoside-related compounds, their safe use (without side effects) should also be highlighted. Thus, the section Conclusion (after the sentence in the lines 161-163) could be completed with the following (or similar) sentence: Their safe use was also investigated, confirming no cytotoxic side effects on mammalian kidney cells (Vero E6) and human erythrocytes (Zürn et al., 2019 Industrial Crops and Products, https://doi.org/10.1016/j.indcrop.2019.111517).

As suggested, the sentence was modified as follows: “In the last decade, interests on verbascoside derivatives has been growing because of the increasing evidence showing their role in the prevention and treatment of various human diseases and disorders [46], and due to their apparent low toxicity in mammals, as observed in non-human primate kidney (Vero E6) cells and human erythrocytes [47]”. The reference suggested by the reviewer was added as [47], and the reference list was consequently modified.

In conclusion the manuscript deserves publication after a minor revision, taking into consideration the above detailed comments.

Reviewer 2 Report

The study on chemical characterization of methanolic extracts of Colebrookea oppositifolia has already reported but this MS has few new information about the identified metabolites. Otherwise, there is no new information in this study. 

The entire experimental section was copied from two main sources, Then I have decided to check the similarity using iThenticate and found an overall 35 % similarity which is not acceptable for publication in IJMS. 

14% similarity from Peron et al 2019, Journal of Pharmaceutical and Biomedical Analysis, Volume 174, Page 663-673

4 % similarity from Bendif et al Journal of Pharmaceutical and Biomedical Analysis, Volume 186, Page 113330 https://www.sciencedirect.com/science/article/pii/S0731708520309298?via%3Dihub#!

The introduction is too general and should be improved, avoid speculations.  Other sections look good and well presented.    I advise authors to reduce the similarity and submit this manuscript as a new submission or resubmission accounting to the journal policy.   

Author Response

The study on chemical characterization of methanolic extracts of Colebrookea oppositifolia has already reported but this MS has few new information about the identified metabolites. Otherwise, there is no new information in this study.

As correctly mentioned by the reviewer, and as we described in our work, characterizations of C. oppositifolia leaves have been already published. However, to the best of our knowledge, only few compounds have been identified up to now, mainly flavonoids (alnustin, chrysin, apigenin, quercetin, for example) and phenolic acids, such as caffeic acid and acteoside. Hence, an exhaustive characterization of the potentially bioactive secondary compounds from this species is still lacking, as well as an evaluation of its anti-inflammatory potential. Furthermore, up to our knowledge, limited information about C. oppositifolia from Nepal is available.

The entire experimental section was copied from two main sources, Then I have decided to check the similarity using iThenticate and found an overall 35 % similarity which is not acceptable for publication in IJMS.

14% similarity from Peron et al 2019, Journal of Pharmaceutical and Biomedical Analysis, Volume 174, Page 663-673

4 % similarity from Bendif et al Journal of Pharmaceutical and Biomedical Analysis, Volume 186, Page 113330 https://www.sciencedirect.com/science/article/pii/S0731708520309298?via%3Dihub#!

Thanks to the reviewer for the comment. All the manuscript was carefully checked and the abovementioned parts were extensively changed to avoid plagiarism. However, the papers from which the parts have been taken were appropriately cited.

The introduction is too general and should be improved, avoid speculations.  Other sections look good and well presented.    I advise authors to reduce the similarity and submit this manuscript as a new submission or resubmission accounting to the journal policy.

Thanks to the Reviewer for the comment. In the introduction, we aimed at giving a complete view on the traditional medicinal knowledge about this plant, as well as on the scientific studies that have been published up to now focusing on its chemical characterization and bioactivity evaluations. All the sources of the information reported have been properly cited.

Following the suggestion of the reviewer, some parts of the introduction that maybe were not very clear were modified, and some more precise information over the chemical characterizations available in literature have been reported.

Reviewer 3 Report

This is an interesting work, however some minor changes are required:

-Discussion should contain more infomation concerning a potential clinical impact of the results.

-Limitations and potential pitfalls of the experiemnt should be also discussed.

-The work presents characterization of metabolites from Colebrookea oppositifolia leaves. Additionally, the authors assessed cytotoxicity and anti-inflammatory and cytotoxic effects. This is an interesting work, however some minor changes are required: -Discussion should contain some infomation concerning a potential clinical impact of the results, e.g. in which disorders, this extract could be useful or clinical contenxt of the aim of the study. -Limitations and potential pitfalls of the experiemnt should be also discussed; especially the culture cell part. Why did the Authors chose the THP1-XBlue™-MD2-CD14  cell line? Also, the potential clinical benefits of inhibition of NF-kB and AP-1 should be briefly explained.

Author Response

This is an interesting work, however some minor changes are required:

-Discussion should contain more infomation concerning a potential clinical impact of the results.

Please see below.

-Limitations and potential pitfalls of the experiemnt should be also discussed.

Please see below.

-The work presents characterization of metabolites from Colebrookea oppositifolia leaves. Additionally, the authors assessed cytotoxicity and anti-inflammatory and cytotoxic effects. This is an interesting work, however some minor changes are required:

-Discussion should contain some infomation concerning a potential clinical impact of the results, e.g. in which disorders, this extract could be useful or clinical contenxt of the aim of the study.

A descriptive sentence was added in the discussion: “If the anti-inflammatory effect of COME is proven in vivo and the particular mechanism of action is elucidated, there will be a therapeutic potential to treat a broad range of inflammatory diseases, at least as food supplement, because of the high popularity of complementary and alternative medicine among general population [67].” 

-Limitations and potential pitfalls of the experiemnt should be also discussed; especially the culture cell part. Why did the Authors chose the THP1-XBlue™-MD2-CD14 cell line? Also, the potential clinical benefits of inhibition of NF-kB and AP-1 should be briefly explained.

THP-1-XBlueTM-MD2-CD14 cell line was used as a model for determination of the effect of COME on the activation of the transcription factors NF-κB and AP-1 because this genetically modified cell line has been specifically designed for testing the activity of these transcription factors. For any further details, please visit the webpage of the manufacturer (https://www.invivogen.com/thp1-blue-nfkb).

On the other hand, the potential clinical benefits of inhibiting NF-kB and AP-1 signaling pathways depend on the fact that, although inflammatory response is regulated at several different levels (gene level, signal-transduction level, and cellular level), the main control point lies at the transcriptional level, as previously demonstrated by other authors (REF: DOI: 10.1038/nri2634). Hence, potential novel anti-inflammatory agents should be directed on the modification of these pathways. A brief motivation was added to the manuscript, as suggested.

Round 2

Reviewer 2 Report

Authors revised manuscript according to the review comments. This can accepted now.